# The PAX Genes: Roles in Development, Cancer, and Other Diseases

**DOI:** 10.3390/cancers16051022

**Published:** 2024-02-29

**Authors:** Taryn Shaw, Frederic G. Barr, Aykut Üren

**Affiliations:** 1Department of Oncology, Lombardi Comprehensive Cancer Center, Georgetown University, Washington, DC 20001, USA; tes66@georgetown.edu; 2Laboratory of Pathology, National Cancer Institute, Bethesda, MD 20892, USA; frederic.barr@nih.gov

**Keywords:** PAX genes, PAX proteins, PAX fusions, developmental disorders, cancer, cancer therapeutics

## Abstract

**Simple Summary:**

Humans possess a group of nine related genes that form the PAX gene family. These genes encode proteins known as the PAX transcription factors, which control gene expression on a large scale and coordinate the development of bodily structures such as the eyes and muscles. The biological functions of PAX genes can be regulated by chemical modifications to their protein structures, interactions with other proteins, and changes in splicing, which can produce different versions of PAX proteins. Mutations in PAX genes can contribute to human diseases such as hypothyroidism, diabetes, and cancer. Further research into the functional consequences of these mutations could uncover novel treatments for these diseases.

**Abstract:**

Since their 1986 discovery in *Drosophila*, Paired box (PAX) genes have been shown to play major roles in the early development of the eye, muscle, skeleton, kidney, and other organs. Consistent with their roles as master regulators of tissue formation, the PAX family members are evolutionarily conserved, regulate large transcriptional networks, and in turn can be regulated by a variety of mechanisms. Losses or mutations in these genes can result in developmental disorders or cancers. The precise mechanisms by which PAX genes control disease pathogenesis are well understood in some cases, but much remains to be explored. A deeper understanding of the biology of these genes, therefore, has the potential to aid in the improvement of disease diagnosis and the development of new treatments.

## 1. Introduction

The discovery of the PAX genes occurred in 1986 with the cloning of the *paired* (*prd*) gene in *Drosophila* [1]. The isolation of PAX gene homologs in many species, including frogs, fish, birds, and mammals, rapidly followed. In humans and other species, the PAX genes perform determining roles in organogenesis. The activity of these transcription factors is tightly regulated; expression is localized to specific cell types and temporally controlled, and mutations contribute to a variety of developmental disorders and cancers. The goal of this review is to provide a broad perspective on PAX gene structure and function and to discuss how the PAX genes may be altered in disease pathogenesis, with the aim of highlighting new therapeutic strategies for targeting PAX family members. References for this work were identified via a PubMed database search with no starting year and ending in January 2024. Search terms used include “paired box”, “PAX”, “paired domain”, “master regulator”, “cancer”, and “development”. Studies were assessed for the use of appropriate controls, rigor, and reproducibility over time in order to be included in this review.

### 1.1. Evolution of PAX-like Genes

Prior to the discovery of the PAX genes, it was known that “homeotic” genes controlled the development of specific body segments. In *Drosophila*, these genes were capable of controlling the development of entire organs or bodily structures and mutating them could cause severe congenital malformations. Additionally, portions of these genes exhibited homology with loci in other metazoans, and this conservation between species supported their crucial role in development [2]. Following the cloning of the *prd* gene in *Drosophila*, it was soon determined that mice possessed a homologous gene, *Pax1*, and that the human genome similarly included homologous genes, such as *PAX1* [3,4]. Today, we know that a wide variety of species possess PAX orthologs, from commonly used laboratory organisms such as *Xenopus*, *Caenorhabditis elegans*, and *Danio rerio* to species used as “ancestral” models such as *Amphimedon queenslandica*, a sponge native to the Great Barrier Reef [5]. Humans possess nine PAX genes, and the corresponding PAX protein sequences can be arranged using amino acid sequence alignments and phylogenetic reconstruction (Figure 1) [6]. The dendrogram presented in Figure 1 divides the PAX genes into four groups, corresponding to structural groups I, II, III, and IV, which are discussed in detail in the following section [6].

All PAX genes have their origins in a proto-PAX ancestor, which likely acquired a paired domain from a Tc1/mariner transposon at least 540 million years ago, around the time of the rapid diversification of metazoan species in the Cambrian explosion [5,10]. It is possible that the acquisition of the paired domain occurred even earlier, as some groups have suggested the existence of PAX-like genes in *Giardia lamblia*, a protozoal intestinal parasite [11]. There are two proposed candidates for the proto-PAX gene based on two putative scenarios for Tc1/mariner transposon acquisition. In one scenario, a locus which included an octapeptide motif and a homeodomain hijacked the paired domain from a Tc1/mariner transposon [10]. This proto-PAX gene is known as *PAXB*-like due to its similarity to the *PAXB* gene found in cnidarians and sponges. In the second scenario, a locus containing the octapeptide motif, homeodomain, and an additional region known as the homeodomain tail hijacked the Tc1 transposon. This proto-PAX gene is known as *PAXD*-like, due to its similarity to the *PAXD* gene in cnidarians. Deciding which scenario is correct would require determining whether the ancestral basal metazoan genome more closely resembled sponges or cnidarians [5]. Among the mammalian PAX genes, Groups I and III are more *PAXD*-like, while groups II and IV are more *PAXB*-like [5]. The large number of human PAX genes likely arose due to gene and genome duplication events following the emergence of the proto-PAX gene.

### 1.2. PAX Protein Structure

The nine genes which make up the human PAX family are transcription factors with common structural elements and can be divided into subgroups based on which of these elements are present (Figure 2). The N-terminal paired domain, which gives the family its name, is present in all nine member genes. It is a DNA-binding domain spanning 128 amino acids and was originally discovered in *Drosophila* as a region of homology between the *prd* and *gsb* loci [1,12,13]. Its secondary structure is arranged in two helix-turn-helix motifs, known as the *PAI* and *RED* subdomains, which facilitate DNA binding (Figure 3) [14,15]. In addition to the paired domain, all PAX proteins also include a C-terminal transactivation domain which acts as a binding site for other transcriptional regulators. The presence or absence of two additional domains defines the PAX subgroup to which each gene belongs. Groups I, II, and III include an eight-amino-acid stretch known as the octapeptide motif, which allows for the binding of proteins capable of down-regulating transcriptional activity [16,17]. Groups II, III, and IV include a second DNA-binding domain, the homeodomain, which is truncated in Group II proteins and has a reduced DNA-binding capacity. Despite this truncation, the residual Group II homeodomain is still capable of participating in some of the same protein–protein interactions seen with the full-length domain [17]. Like the paired domain, the full-length homeodomain includes a helix-turn-helix structure that facilitates binding to DNA. This region typically binds DNA as a dimer, although it can also cooperate with the paired domain to bind certain nearby recognition elements [18,19].

In combination, the various structural domains in PAX proteins help give rise to the unique developmental programs driven by the different family members. Post-translational modifications, alternative splicing, tissue-specific and temporal expression changes, and protein–protein interactions add further layers of complexity to the differences between PAX family members, leading to a wide range of phenotypes during embryogenesis.

## 2. Regulation of PAX Family Gene Expression

The activity of the PAX transcription factors can be regulated by post-translational modifications, interactions with partner proteins, and degradation. At the transcript level, PAX gene translation can be regulated by miRNAs or alternative splicing, which generates a wide range of PAX isoforms. Although they frequently act as master regulators, the transcription of PAX genes can also be regulated by a number of upstream transcription factors, as well as via autoregulatory loops. These regulatory mechanisms combine to influence each other and the functions of each PAX gene. Although it would be impossible to detail all the possible mechanisms of PAX gene regulation, examples will be provided of key regulatory processes that are necessary for normal development, disrupted in disease, or potentially targetable for therapeutic purposes.

### 2.1. Upstream Transcription Factors

PAX expression can be driven by a number of other transcription factors, including other PAX genes. As an example, the Group II PAX genes (*PAX2*, *PAX5*, and *PAX8*) contribute to the development of the central nervous system (CNS). In the development of the midbrain–hindbrain boundary, the transcription factors OCT3 and OCT4 trigger the transcription of *PAX2* in the neural plate during gastrulation. PAX2 then initiates the transcription of *PAX5* and *PAX8*, which trigger the expression of other genes that are both necessary for the specification of the midbrain–hindbrain boundary as well as for maintaining the continued expression of *PAX2/5/8* [21]. This example highlights the complexity of transcriptional regulatory networks surrounding the PAX genes in development. The transcription of a PAX gene can be initiated by other transcription factors, which can trigger the transcription of different PAX genes and downstream genes and can become self-sustaining.

### 2.2. miRNAs

The timing of PAX gene expression during development is also regulated by various miRNAs with tissue-specific expression patterns. For example, the miRNAs mi-R1 and miR-206 are expressed in the myotome as it differentiates into skeletal muscle and reduce PAX3 expression over the course of muscle development [22]. The expression of these miRNAs may also be deregulated in disease states, altering PAX gene expression levels accordingly. In rhabdomyosarcomas, mi-R1 and mi-R206 are expressed at lower levels than in normal skeletal muscle [23]. The ectopic expression of these miRNAs can decrease the expression of the PAX3 protein in embryonal rhabdomyosarcoma. However, these miRNAs are incapable of reducing the expression of the oncogenic fusion protein PAX3::FOXO1 in alveolar rhabdomyosarcoma, as the fusion protein lacks the 3′UTR that would normally be the target for these miRNAs [23].

### 2.3. Alternative Splicing and PAX Isoforms

Alternative splicing is a post-transcriptional mechanism for generating multiple transcript variants from a single transcribed gene. These variants may be translated into structurally identical proteins, but they may also generate different protein isoforms with altered structural domains and functions. Another mechanism for the generation of protein isoforms is the use of alternative transcription start sites, which generate similar but distinct mRNA transcripts. Eight of the nine human PAX genes undergo alternative splicing or utilize alternative transcription start sites to produce more than one transcript, with as few as two resulting isoforms (*PAX1* and *PAX4*) and as many as fifteen (*PAX6*) (Table 1). These isoforms differ not only in their structural domains, but in their potential protein–protein and protein–DNA interactions. Because not all reported PAX isoforms are detectable in adult human tissues, it is likely that some of these isoforms play a temporally restricted role in embryonic development [24]. Multiple alternatively spliced isoforms may also be present at the same time, as is the case for PAX3 and PAX7 in the developing embryo and in rhabdomyosarcoma tumors [25,26]. Alternative splicing frequently results in alterations to the C-terminal region containing the transactivation domain [24].

**Table 2 cancers-16-01022-t002:** PAX genes and associated developmental processes. This table summarizes the reported key developmental roles and associated developmental disorders of the nine PAX genes. Associated disorders are listed using OMIM phenotype names. Relevant references are cited in the text. The roles of these genes in human cancers are summarized in Table 3.

Subgroup	Gene	Aliases	Developmental Role(s)	Associated Disorder(s)
Group I	*PAX1*	HuP48	Axial and appendicular skeleton, thymus, parathyroid gland	Otofaciocervical syndrome-2 (OTFCS2)
	*PAX9*	-	Axial and craniofacial skeleton, teeth	Selective tooth agenesis-3 (STHAG3)
Group II	*PAX2*	-	Kidney, CNS	Papillorenal syndrome (PAPRS), focal segmental glomerulosclerosis-7 (FSGS7)
	*PAX5*	BSAP	CNS, B cells	-
	*PAX8*	-	Kidney, CNS, thyroid	Congenital hypothyroidism
Group III	*PAX3*	HuP2	Skeletal muscle, neural crest, CNS	Waardenburg syndrome (Types 1 and 3), craniofacial-deafness-hand syndrome (CDHS)
	*PAX7*	HuP1, RMS2	Skeletal muscle, neural crest, CNS	Congenital myopathy-19 (CMYP19)
Group IV	*PAX4*	-	GI endocrine cells	Maturity-onset diabetes of the young (Type 9, MODY9), diabetes mellitus (Type 2), ketosis-prone diabetes
	*PAX6*	MGDA, WAGR	GI endocrine cells, eye, CNS	Aniridia, anterior segment dysgenesis 5 (ASGD5), foveal hypoplasia 1 (FVH1), keratitis

While the roles of each isoform have yet to be fully elucidated, there are several examples of isoform-specific PAX functions. For example, some PAX isoforms may exhibit a dominant negative effect on others, as in the case with PAX3. When generated, the PAX3f isoform has the ability to reduce transcription by the PAX3a isoform by occupying its typical binding sites without promoting transcription at those loci. This occurs because the PAX3f isoform lacks most of its C-terminal transcriptional activation domain [27]. This suggests that a change in the balance of isoform percentages could have major consequences for development and disease.

Another example of isoform-specific function is found in *PAX6*, where two isoforms must cooperate in order to promote normal corneal development via the expression of different keratin genes. PAX6a is capable of activating *KRT4* transcription in coordination with KLF4, while PAX6b (in coordination with both KLF4 and OCT4) is capable of activating *KRT12* transcription [28]. The activity and binding partners of both isoforms differ, yet both isoforms contribute to the development of the corneal epithelium.

### 2.4. Post-Translational Modifications

Phosphorylation sites have been reported for all nine PAX proteins, and acetylation, ubiquitylation, methylation, sumoylation, and caspase cleavage have also been reported among different sets of PAX family members. In some cases, the functional outcomes of these modifications are unclear, while others have defined consequences. One such example is PAX3 phosphorylation at Ser205, a residue in the region surrounding the octapeptide motif (Figure 2). This phosphorylation event is temporally restricted; in proliferating mouse myoblasts, there is a high level of phosphorylation at this residue, but phosphorylation is lost in cells that begin to differentiate [29]. Interestingly, it has been shown that in rhabdomyosarcomas which express the PAX3::FOXO1 fusion protein, Ser205 remains phosphorylated even as myogenic differentiation is triggered [30]. A similar phosphorylation event occurs at Ser203 in the PAX7 protein, another Group III PAX family member fused to FOXO1 in a subset of alveolar rhabdomyosarcoma. 

PAX protein residues may also be modified via redox reactions. In particular, PAX8 has three cysteine residues in its paired domain that are capable of being glutathionylated. The oxidation of two of these cysteine residues using diamide treatment lowers the level of DNA binding that occurs via the paired domain [31]. The DNA binding capability of the paired domain can be restored by treatment with a reducing agent such as dithiothreitol [31]. Additionally, the reduction of the glutathionylated residues by the redox factor APEX1 can restore the DNA-binding capabilities of oxidized PAX8 [31]. 

### 2.5. Protein–Protein Interactions

The PAX proteins interact with a number of other proteins, primarily other transcription factors. Additionally, PAX proteins can form homo- or heterodimers with other PAX proteins in order to bind DNA via their homeodomains. For example, the *PAX3* and *PAX7* proteins exhibit overlapping expression patterns in developing embryos, and can come together to bind DNA and facilitate myogenesis [32].

One example of an interaction between PAX proteins and other transcription factors is the interaction between PAX5 and ETS1. ETS1, an Ets family member, plays a variety of roles in tissue development and is critical for B cell function. ETS1 promotes the expression of its target gene *CD79A*, which encodes a subunit of the B cell antigen receptor. In order for ETS1 to bind at the recognition sequence of this gene, it must form a complex with PAX5 [33]. The resulting ternary complex between PAX5, ETS1, and DNA facilitates the expression of *CD79A.*

Interactions between PAX proteins and their binding partners can also influence the state of chromatin. The binding of PAX2 to PTIP increases transactivation by PAX2, as well as maintaining an open chromatin state [34]. The opening of chromatin occurs because the interaction between PAX2 and PTIP recruits MLL family histone methyltransferases, leading to trimethylation at lysine 4 of histone H3 (H3K4me3), which promotes chromatin relaxation [34]. 

## 3. PAX Genes during Development

All nine PAX transcription factors have been shown to play fundamental roles in the development of specific tissue types (Table 2, Figure 4). For this reason, they are often known as “master regulators”. Master regulators can be defined by several key characteristics: high expression levels in the relevant tissue type, enhancing their own expression through positive feedback loops, high binding at the regulatory elements of actively expressed tissue-specific genes, and promoting the transcription of lineage-specifying gene groups while inhibiting the transcription of lineage-inappropriate gene groups [35]. The control of specific gene groups is often established via interactions with coactivators such as p300 and Mediator complex members, as well as corepressors, such as genes belonging to the Polycomb family [35]. One non-PAX transcription factor that exemplifies the characteristics of a master regulator is *NKX2-5*, which governs the development of the heart. The homolog of this gene was first discovered in *Drosophila* and was named tinman, because the disruption of the gene prevented the specification of the heart in developing embryos [36]. In mice, *Nkx2-5* transcripts are highly expressed in myocardiogenic progenitor cells and continue to be expressed over the course of development into adult cardiomyocytes [37]. It has been shown that NKX2-5 is capable of binding to its own regulatory regions [38]. *Nkx2-5* is also predicted to regulate the transcription of hundreds of genes, including dozens of genes involved in heart development [39]. These characteristics are also exhibited to various degrees by the proteins encoded by the PAX genes. For example, *PAX6* is expressed at high levels in the developing murine eye, where it binds regulatory regions for a set of genes that are critical for lens development [40,41]. In mice, it has been shown that *PAX6* controls an autoregulatory loop to maintain its own transcription [42]. In *Drosophila*, Pax6 cooperates with Polycomb group (PcG) proteins to suppress the reprogramming of the eye, preventing an eye–wing transition [43]. Below, the role of *PAX6* in eye development will be discussed in more detail, along with the roles of other PAX genes in the specification of various tissue types. Many insights into the developmental roles of Pax genes were gained via the use of animal models, particularly mouse models. Where relevant, the “master regulator” characteristics that drive these developmental programs will be emphasized. 

### 3.1. PAX1 and PAX9 in Skeletal Development

The Group I PAX genes play crucial roles in skeletal development, particularly in the formation of the axial and craniofacial skeleton and teeth. Both *Pax1* and *Pax9* are highly expressed in the somites and sclerotome of the mouse embryo, but this expression pattern is lost in the fully developed vertebral column of adult mice [3,44]. Loss-of-function studies in mice have also demonstrated that despite their structural similarities, these genes do not play entirely redundant roles. In mice, there are three naturally occurring *Pax1* mutants: *undulated* (*un*), *undulated extensive* (*un^ex^*), and *undulated short-tail* (*Un^s^*) alleles [45]. Of these, the *un* and *un^ex^* mutations are considered less severe hypomorphs, which result in axial skeletal defects in homozygous mice. The *Un^s^* phenotype arises from the deletion of the entire *Pax1* locus and is more severe; heterozygous mice exhibit clear skeletal abnormalities including a short, kinked tail, and homozygous mice have severe skeletal malformations that lead to perinatal mortality [45]. Interestingly, the phenotype exhibited by *Un^s^* mice is more severe than that seen in *Pax1*-null mutants created by gene targeting. The homozygous *Pax1*-null mutant phenotype is similar to *Un*^s^ heterozygotes, and heterozygous *Pax1*-null mice are indistinguishable from wild-type mice [46]. One explanation for this observation is that *Un^s^ Pax1* deletion also leads to the increased expression of *Nkx2-2* in the sclerotome, possibly via interactions between the *Pax1* enhancers and the *Nkx2-2* promoter [47].

In mice, *Pax9* also plays a major role in vertebral column development that is somewhat distinct from that of *Pax1*. Mouse *Pax1^−/−^;Pax9^−/−^* double homozygous mutants exhibit a phenotype that is far more severe than either homozygous *Pax^−/−^* mutant [44,48]. While *Pax9^−/−^* mice do not exhibit obvious defects in vertebral column formation, this homozygous mutation is often lethal due to defects in craniofacial formation, highlighting the unique role that *Pax9* plays in the genesis of the palate and teeth [49]. In line with its role as a master regulator in humans, *PAX9* has been demonstrated to control the expression of a large set of genes belonging to the *Wnt/β*-catenin, *Osr2*, and *TGFβ*3 pathways, which help control palatogenesis [50]. Together, the Group I PAX genes play crucial roles in skeletal development that are both overlapping and divergent.

### 3.2. PAX2 and PAX8 in Renal Development

In mice, *Pax2* is highly expressed in the developing urinary excretory system but is undetectable in adult fully formed kidneys and urogenital tracts [51]. The transgenic Tg8052 line includes a heterozygous deletion at the *Pax2* locus, and these mice exhibit multiple kidney abnormalities, including aplasia and hypoplasia [52]. In *Pax2^−/−^* mice generated by homologous recombination, there is a complete absence of kidney, ureter, and genital formation [53].

In *Xenopus*, *pax8* is strongly expressed early in kidney development, and when Pax8 protein levels are depleted using translation-blocking morpholinos, the pronephric tubule fails to form [54,55]. *Pax8^−/−^* mice do not exhibit defects in renal development, but they die shortly after weaning due to defects in thyroid formation [56]. *Pax2^+/−^;Pax8^+/−^* mice develop kidneys that are more severely hypodysplastic than those of *Pax2^+/−^* mice [57]. This suggests that in *Pax8^−/−^* mice, two functional *Pax2* alleles may be completely capable of directing kidney formation. In *Pax2^+/−^* mice, a single functional *Pax8* allele can partially compensate for the loss of a *Pax2* allele in kidney formation.

The *Pax2*- and *Pax8*-mediated control of gene expression in the renal system is established in part due to interactions with epigenetic coregulators. The proteins encoded by Group II Pax genes have been shown to interact with corepressors in the Groucho family, particularly GRG4 [17]. The interaction between PAX2 and GRG4 triggers H3 lysine 27 trimethylation (H3K27me3), which leads to chromatin condensation and transcriptional repression [58]. This transcriptional repression can be overcome via interactions between phosphorylated PAX2 and the coactivators such as PTIP and members of the Trithorax-group methyltransferases [34]. These interactions lead to H3 lysine 4 trimethylation (H3K4me3), which promotes chromatin relaxation and transcriptional activation. These specific epigenetic mechanisms are required to control the kidney-specific transcriptional program [59]. Overall, *Pax2* and *Pax8* play somewhat redundant roles in kidney formation, but correct gene dosage is critical for normal development.

### 3.3. PAX5 in B Cells

The formation of B cells from hematopoietic stem cells requires several steps. Prior to B cell lineage commitment, hematopoietic stem cells exist as pre-pro B cells, which express markers such as *B220*, a *CD45* isoform [60]. Commitment to the B cell lineage requires *PAX5* expression, which drives the transcription of many target genes, including the marker *CD19*, as well as the expression of *IL-7*. In pre-pro B cells, the immunoglobulin heavy (*IgH*) locus must undergo V(D)J recombination in order to form the pre-B cell receptor (pre-*BCR)* and continue to the pre-B cell stage. Pre-B cells then require the recombination of the immunoglobulin light (*Igl*) chains to form the VJ complex and mature B cell receptor (*BCR*) [60]. *PAX5* and *IL-7* must be expressed throughout these processes for proper B cell development to occur. While the *Pax5^−/−^* mutation is not embryonic lethal in mice, most mice with this homozygous deletion die perinatally, and none are capable of producing any of the cells in the B lymphocyte lineage [61]. Because *Pax5^−/−^* hematopoietic stem cells are incapable of progressing to B cell commitment, they can be used to rescue T cell, but not B cell, formation in *Rag2^−/−^* mice that lack the ability to form T or B lymphocytes [62]. 

In the specification of the B-lymphocyte lineage, PAX5 is a prime example of a master regulator. It is expressed at high levels in the hematopoietic progenitors that are in the pro-B cell stage and continues to be expressed through the subsequent stages of B-cell differentiation [63]. It binds at the regulatory elements of many genes required for specifying the B-cell state, activating the transcription of this group of genes in part via interactions with the coactivator p300 [64]. As a member of the Group II PAX genes, PAX5 also interacts with Groucho family corepressors to inhibit the transcription of genes that specify other cell lineages [17]. 

### 3.4. PAX8 in Thyroid Development

There are two cell lineages that comprise the thyroid: follicular cells, which produce thyroxine, and parafollicular C cells, which produce calcitonin. While it was recently determined that both cell types arise from an endodermal population of thyroid precursor cells that express *PAX8*, only follicular cells have been shown to require *PAX8* for complete differentiation [56,65]. In mice, *Pax8* is highly expressed in the developing thyroid gland, and *Pax8^−/−^* mice fail to develop a thyroid past the bud stage [56,66]. *Pax8* and *Nkx2-1* can also be ectopically expressed in mouse embryonic stem cells to trigger differentiation into thyroid follicular cells [67]. Consistent with its role as a master regulator, *PAX8* also drives the expression of many target genes that play key roles in thyroid differentiation, including thyroglobulin, thyroperoxidase, and sodium/iodide symporter (*NIS*) genes [68]. 

### 3.5. PAX3 and PAX7 in Myogenesis

During development, the cells that form the paraxial mesoderm give rise to the somites, which eventually become the sclerotome and dermomyotome, which gives rise to the myoblasts that form the limb muscles. In mice, *Pax3* is expressed in the somites and throughout the dermomyotome, and continues to be expressed through the development of the myoblasts, including during their migration to the developing limbs [69]. In the homozygous *splotch* (*Sp*) mutant mouse model, which involves mutations in the *Pax3* homeodomain, skeletal muscle development is severely reduced, specifically in the shoulders and limbs [70]. Several traditional markers for skeletal muscle, such as *Myog* and *Myf5*, are absent, and the migrating myoblasts that eventually form the limb muscles are also absent [69]. Additionally, although the skeletal muscle marker *MyoD* can be activated in the somites of *Sp* mice, these cells die rapidly, indicating that Pax3 is also required to inhibit programmed cell death in this cell population [71]. Homozygous *Sp* mutants also die mid-gestation, although this lethality is due to defects in neural crest formation, which are discussed below.

In *Sp* mice, the loss of *Pax3* causes *Pax7* to be compensatorily expressed in somitic progenitor cells [71]. *Pax7^−/−^* mutant mice are much smaller than their wild-type counterparts, have low muscle mass, and die within 2–3 weeks of birth. Unlike *Sp* mutant mice, *Pax7^−/−^* mutants still express the skeletal muscle markers *MyoD* and *Myf5* [72]. However, *Pax7^−/−^* mice do not form satellite cells, the population of cells that gives rise to differentiated muscle cells during muscle growth or following injury or disease in adult animals [72]. In line with the complementary but distinct roles of *PAX3* and *PAX7* in myogenesis, double knockout *Pax3^−/−^;Pax7^−/−^* mice exhibit profound defects in somite formation and a loss of muscle mass that is more severe than the phenotype for either of the separate homozygous mutations [73].

### 3.6. PAX3 and PAX7 in Neural Crest Formation

The neural crest is a collection of cells that arises from the embryonic ectoderm following neuroectodermal tissue differentiation and neurulation [74]. These cells initially reside in the area between the surface ectoderm and the neural tube and undergo an epithelial to mesenchymal transition in order to migrate and form a variety of cell types, including the ganglia, Schwann cells, smooth muscle, and melanocytes [74]. In mice, *Pax3* is expressed in the developing neural crest cells and continues to be expressed through their migration and differentiation into the cells of the dorsal root ganglion [75]. As mentioned above, the homozygous *Sp* mouse model involves mutations in the *Pax3* gene. Accordingly, *Sp* mice exhibit defects in neural crest cell migration and neural tube closure, leading to the failure to form enteric ganglia and congenital heart disease [76,77]. These defects lead to mortality during gestation.

The neural crest phenotype observed in *Pax7^−/−^* mice is less severe; mice survive gestation, but die after weaning [78]. The reduced severity of this phenotype is likely due to its restriction to the cephalic neural crest cells, which form the craniofacial neurons, glia, cartilage, and connective tissues. For this reason, *Pax7^−^*^/*−*^ mice exhibit malformed maxillae, serous glands, and nasal capsules, but do not exhibit defects in cardiac or ganglial development [78]. As seen in myogenesis, the neural crest defects observed due to the loss of either *Pax3* or *Pax7* may be mitigated in part by the upregulation of the other remaining gene, as the *Pax3^−/−^;Pax7^−/−^* phenotype is far more severe than the loss of either single gene.

### 3.7. PAX4 and PAX6 in Pancreatic Endocrine Cells

The pancreas contains five different types of endocrine cells organized into the pancreatic islets: alpha cells, which produce glucagon; beta cells, which produce insulin; delta cells, which produce somatostatin; PP cells, which produce pancreatic polypeptide; and epsilon cells, which produce ghrelin. Endocrine precursor cells require *PAX4* and *PAX6* expression to properly differentiate into all five types of pancreatic endocrine cells, although these are not the only required factors for GI endocrine cell development. In *Pax4^−/−^* mice, beta and delta cells do not form, although alpha and epsilon cell formation is higher [79,80]. Because these mice fail to form beta cells, they die shortly after birth due to hyperglycemia. In *Pax6^−/−^* mice, alpha cells do not form, and beta, delta, and PP cells do not properly organize into pancreatic islets [81]. *Pax4^−/−^;Pax6^−/−^* mice fail to form any adult pancreatic endocrine cells [81]. Interestingly, *Pax6* continues to be expressed in all adult pancreatic endocrine cells. In beta cells, it appears to maintain the active expression of beta cell-specific genes while suppressing the expression of genes related to other endocrine lineages [82].

### 3.8. PAX6 in Ocular Development

The eye is a complicated structure; a functioning human eye includes the cornea and lens, which arise from the surface ectoderm, as well as the retina, iris, and ciliary body, which arise from the neural plate [83]. Despite this complexity, the development of the eye is coordinated in large part by a single gene, *PAX6*, making it one of the most striking examples of a PAX gene master regulator. As mentioned above, *PAX6* represents the major characteristics of a master regulator, including high expression in the developing tissue type, maintaining its own expression via autoregulatory loops, occupying the regulatory regions of a large number of genes, and activating the transcription of lineage-specifying genes while repressing the transcription of lineage-inappropriate genes [40,41,42,43]. An eyeless mutant was first reported in *Drosophila* over a century ago, long before the discovery of the *prd* gene [84]. Later, it was determined that the locus responsible for the *Drosophila* eyeless phenotype, called *ey*, exhibited homology to the *Pax6* gene in mice (called *small-eye*) and to human *PAX6* [85]. This discovery indicated that mammalian and insect eye development shared a common genetic mechanism despite major differences in eye structure and function between these two classes.

In mice, *Pax6* is expressed throughout the course of eye development, first appearing in the optic sulcus and continuing to be expressed as the optic sulcus develops into the optic vesicle, which eventually becomes the optic cup and lens placode [40]. The homozygous *Pax6 small-eye* (*Sey*) mutation results in a complete lack of eye formation, and heterozygous mice have much smaller lenses than their wild-type counterparts [86]. The homozygous *Sey* mutation also leads to perinatal mortality, as these mice fail to develop nasal pits and newborn mice cannot breathe through their mouths [86]. Many other variant *Pax6* alleles have been reported in mice, but one of the most severe, *Pax6^3Neu^*, arises from the insertion of an alanine, which leads to a truncation of the protein after the paired domain [87]. Other mutations can alter *Pax6* alternative splicing, generating a greater proportion of a PAX6 isoform that primarily binds to DNA via the *RED* subdomain of the paired domain, and altering the gene regulatory regions which are bound by PAX6 [88]. 

## 4. PAX Genes in Human Disease

Because of the critical roles that the PAX genes play in human development, mutations in these genes can give rise to a wide variety of associated human genetic disorders (Table 2). The severity of the disease phenotype often corresponds to the severity of the structural disturbance observed at the protein level and/or whether the mutation is homozygous or heterozygous, as many PAX phenotypes are dependent on gene dosage. Below, these genetic disorders will be discussed, along with their modes of inheritance and the observed structural and functional consequences of their associated PAX mutations.

### 4.1. PAX1 in Otofaciocervical Syndrome

As in the *undulated* mouse models, alterations in human *PAX1* are linked to genetic disorders involving skeletal anomalies. For example, otofaciocervical syndrome (OTFCS) is inherited in an autosomal recessive pattern, and can arise as a result of homozygous *PAX1* mutations [89]. Symptoms of this syndrome include skeletal anomalies such as winged scapulae, low-set clavicles, and short stature, as well as low-set, cup-shaped ears, preauricular pits, hearing loss, T cell deficiency, and mild intellectual disability. Several studies of consanguineous families with members affected by OTFCS have revealed a variety of homozygous *PAX1* mutations, including a missense mutation in the paired box region which contributes to reduced DNA-binding capacity, a nonsense mutation predicted to trigger nonsense-mediated decay, and an insertion leading to a truncating frameshift mutation [89,90,91]. 

### 4.2. PAX9 in Tooth Agenesis

Mutations in *PAX9* have been associated with autosomal dominant tooth agenesis in multiple families [92,93,94,95,96]. The severity of tooth agenesis can range from hypodontia (agenesis of fewer than six teeth) to oligodontia (agenesis of six or more teeth). Frameshift and missense mutations in the *PAX9* paired domain have been linked to oligodontia, while a missense mutation and an insertion leading to a truncation have been reported in families with hypodontia [92,93,94,95,96]. Tooth agenesis severity appears to be correlated to the effects of these mutations on the DNA-binding capability of the PAX9 protein. Certain mutations such as K19E and G51S allow PAX9 to retain a degree of binding to paired box recognition sequences, and these mutations are associated with milder tooth agenesis, while mutations such as R26W and L21P render the protein incapable of binding to DNA, and are associated with a more severe phenotype [97].

### 4.3. PAX2 in Renal and Ocular Disorders

*PAX2* mutations have been linked to a range of kidney and eye malformations. In particular, an autosomal dominant form of papillorenal syndrome (PAPRS) has been linked to heterozygous frameshift, splice site, and nonsense mutations in *PAX2* [98,99,100]. These mutations are often known or predicted to result in a truncated version of the PAX2 protein, which lacks some or all of the C-terminal transactivation domain [101]. Missense or in-frame insertions linked to PAPRS occur in the paired domain [101]. PAPRS is characterized by renal hypoplasia, which often leads to end-stage renal disease, and optic nerve colobomas, sometimes referred to as morning glory disc anomalies [98]. Another disorder linked to heterozygous *PAX2* mutations is focal segmental glomerulosclerosis-7 (FSGS7) [101]. FSGS7 is characterized by the scarring of some of the kidney’s glomeruli, leading to proteinuria and occasionally end-stage renal disease. It is commonly considered less severe than PAPRS and is often caused by heterozygous missense mutations in the *PAX2* paired domain that reduce the protein’s DNA-binding capability. More severe cases of FSGS7 are associated with heterozygous *PAX2* nonsense mutations [101].

### 4.4. PAX8 in Congenital Hypothyroidism

Congenital hypothyroidism is typically associated with thyroid dysgenesis, which can include the absence, incorrect localization, or reduced size of the thyroid gland [102]. Heterozygous missense mutations localized to the *PAX8* paired domain are capable of causing congenital hypothyroidism, and it has been demonstrated that the L62R mutation in the paired domain leads to a reduction in PAX8 DNA binding [102,103,104]. Interestingly, in at least one case, congenital hypothyroidism caused by a *PAX8* mutation was also associated with unilateral kidney agenesis, highlighting the role that this gene also plays in renal development [103].

### 4.5. PAX3 in Waardenburg and Craniofacial-Deafness-Hand Syndromes

Mutations in human *PAX3* are associated with Waardenburg syndrome (WS), which is commonly divided into four separate types based on observed symptoms and causative genetic variants [105]. Both type 1 and type 3 WS (WS1 and WS3, respectively) have been associated with *PAX3* mutations [106,107,108]. WS1 is characterized by congenital hearing loss, pigmentary anomalies in the skin, eyes, and hair, including a white forelock, and dystopia canthorum, an increased distance between the inner corners of the eyes. In WS1, inheritance is autosomal dominant, and a variety of heterozygous *PAX3* mutations have been described [106,107,109]. WS3 is characterized by the same symptoms listed for WS1, in addition to upper limb abnormalities. Unlike WS1, WS3 can be inherited in either an autosomal dominant or an autosomal recessive manner [108,110]. Many WS *PAX3* mutations are missense mutations that occur in the paired domain and affect the ability of the PAX3 protein to bind to DNA [111]. These mutations may also affect DNA binding by the homeodomain, highlighting the coordination between the paired domain and homeodomain that may be required to bind certain sequences [111].

*PAX3* mutations have also been described in craniofacial-deafness-hand syndrome (CDHS), a disorder with symptoms that include hand abnormalities, small or absent wrist and nasal bones, hearing loss, and other facial anomalies [112]. CDHS is inherited in an autosomal dominant manner, and a heterozygous N47K missense mutation in the paired domain has been reported in patients with CDHS [113].

### 4.6. PAX7 in Congenital Myopathy

Homozygous mutations in *PAX7* have recently been reported in several families with congenital myopathy [114]. Congenital myopathy is associated with progressive muscle weakness and atrophy, joint contracture, dysmorphic facial features, scoliosis, and difficulty walking. As mentioned above, *Pax7^−/−^* mice lack myosatellite cells. In human patients with biallelic *PAX7* mutations, the satellite cell pool is exhausted [114]. The reported mutations included two nonsense mutations, a splice site mutation, and a missense mutation. The most severely affected patient had a *PAX7* nonsense mutation and a complete lack of *PAX7* expression, suggesting nonsense-mediated decay. The reported missense mutation, R56C, fell within the protein’s paired domain, and a computer simulation indicated that this amino acid substitution would be likely to reduce the protein’s DNA-binding ability [114].

### 4.7. PAX4 in Diabetes

*PAX4* mutations contribute to several types of diabetes. In a cohort of over 300 Japanese subjects, heterozygous and homozygous R121W mutations in *PAX4* were found in 2% of patients with type 2 diabetes, while this mutation was not present in any unaffected subjects [115].

*PAX4* missense mutations have also been reported in patients with maturity onset diabetes of the young (MODY), a type of diabetes that resembles adult-onset type 2 diabetes but occurs in juvenile patients. One reported mutation, R164W, occurs in the homeodomain and reduces the ability of the PAX4 protein to regulate activity at the insulin and glucagon promoters [116].

Ketosis-prone diabetes (KPD) has also been linked to *PAX4* mutations [117]. KPD is a subtype of type 2 diabetes that shares some of the characteristics of both type 1 and type 2 diabetes, including susceptibility to diabetic ketoacidosis, as seen in type 1 diabetes, as well as later onset, as seen in type 2 diabetes. It can be either insulin-dependent or independent, fluctuating over time. KPD has specifically been associated with homozygous missense mutations in *PAX4* including R133W and R37W, found in West African populations [117].

### 4.8. PAX6 in Ocular Disorders

*Pax6* has been demonstrated to play a role in eye development in mouse models such as *Sey*, and *PAX6* mutations have also been reported in many different human eye malformations. The most well-studied of these is aniridia, a congenital absence of the iris. A majority of aniridia cases are connected to a *PAX6* mutation; a 2008 study found that of 125 patients with aniridia, 94% had a detectable mutation at the *PAX6* genomic locus [118]. Most of these mutations are heterozygous, and insertions, deletions, missense, and nonsense mutations have all been reported across many PAX6 structural domains [88,119,120]. In one case, a patient with biallelic *PAX6* mutations from two parents with aniridia was born with microphthalmia, an abnormally small size of one or both eyes [121].

Heterozygous *PAX6* mutations have also been reported in patients with anterior segment dysgenesis, a disorder that involves multiple eye malformations, including Peters anomaly, which causes the clouding of the cornea and visual impairment and congenital cataracts. Several missense *PAX6* mutations have been associated with this disease, including P375Q, which reduces DNA binding by the paired domain, and Q422R, which reduces DNA binding by the homeodomain [119]. Autosomal dominant *PAX6* mutations have also been reported in foveal hypoplasia 1 and keratitis [122,123].

## 5. PAX Genes and Cancer

Given the ability of the PAX genes to regulate the transcription of large gene networks, it is unsurprising that alterations in the PAX genes have been associated with human cancers (Table 3). However, the overexpression of PAX genes in model systems is typically insufficient for inducing cancer [124]. Therefore, cellular and tissue context play important roles in PAX-related pathogenesis in cancer.

**Table 3 cancers-16-01022-t003:** PAX genes in human tumors. This table highlights some of the known roles of PAX genes in human cancers but is not exhaustive.

Gene	Associated Human Cancer	Observation	Reference
*PAX1*	Colorectal carcinoma	Greater methylation at the *PAX1* promoter and lower mRNA and protein expression levels of PAX1 in colorectal cancer vs. paired normal tissue samples	[125]
	Cervical cancer	Greater methylation at the *PAX1* promoter in patients with high-grade lesions as compared to patients with lower-grade lesions or normal cervical tissue	[126]
*PAX9*	Esophageal carcinoma	Inverse correlation between PAX9 protein expression level and malignancy of epithelial lesions	[127]
*PAX2*	Ovarian cancer	Increased expression of *PAX2* mRNA in low-grade and high-grade carcinoma samples as compared to normal ovarian surface epithelia. Increased PAX2 protein expression in low-grade carcinoma samples, compared to no PAX2 protein expression in high-grade carcinomas or normal ovarian surface epithelia	[128]
	Wilms tumor	Higher *PAX2* expression in Wilms tumor samples compared to normal adult kidney	[129,130]
*PAX5*	Non-Hodgkin lymphoma	A t(9;14)(p13;q32) chromosomal translocation generating the PAX5::IGH fusion gene, which brings the enhancer of the *IGH* gene in close proximity to the *PAX5* gene and increases *PAX5* transcription	[131,132,133]
	Acute lymphoblastic leukemia	PAX5::ELN, PAX5::ETV6, PAX5::FOXP1, PAX5::PML gene fusions	[134,135,136,137]
*PAX8*	Follicular thyroid carcinoma	A t(2;3)(q13;p25) chromosomal translocation generating the PAX8::PPARg fusion protein, which acts as a dominant negative inhibitor of wild-type PPARg and activates transcription of some PAX8 and PPARg target genes	[138,139]
	Wilms tumor	Higher *PAX8* expression in Wilms tumor samples compared to normal adult kidney	[140]
*PAX3*	Alveolar rhabdomyosarcoma	A t(2;13)(q35;q14) chromosomal translocation generating the PAX3::FOXO1 fusion protein, which has a 10- to 100-fold increase in transcriptional activity compared to PAX3 at regulatory sites of target gene transcription	[141,142]
	Melanoma	Increased expression of *PAX3* in cutaneous melanoma as compared to benign lesions or normal skin	[143]
*PAX7*	Alveolar rhabdomyosarcoma	A t(1;13)(p36;q14) chromosomal translocation generating the PAX7::FOXO1 fusion protein	[144]
*PAX4*	Insulinomas	Increased expression of *PAX4* in insulinoma samples as compared to normal islets	[145]
*PAX6*	Pancreatic carcinoma	Increased expression of PAX6 in pancreatic carcinoma tumors and cell lines as compared to normal adult pancreatic exocrine cells	[146]
	Glioblastoma	Inverse correlation between PAX6 protein expression level and malignancy of astrocytic gliomas	[147]

Perhaps the most devastating oncogenic effects from PAX genes arise as a result of chromosomal translocations. The resulting chimeric genes often juxtapose the paired domain of the PAX gene with the transactivation domain of another transcription factor and can alter transcriptional programs to favor a stem cell-like program of self-renewal. While the knowledge of PAX gene involvement in human cancers is continually evolving, current understanding may help to inform the design of new therapeutic options for PAX-related cancers.

### 5.1. PAX Gene Expression in Cancer

Wild-type PAX gene expression is detected in a number of human cancers and is typically associated with negative outcomes. *PAX3* and *PAX7* are expressed in melanomas and sarcomas, and have been associated with tumor-promoting activity [143]. In melanomas as well as in benign melanocytic nevi, *PAX3* expression can be detected and has been used as a biomarker [148]. The exact mechanism by which *PAX3* might contribute to melanomagenesis is not fully understood. Early studies indicated that PAX3 was capable of transactivating the promoter of *MITF*, a transcription factor involved in melanocyte differentiation and an oncogene associated with melanoma cell survival [149]. More recent studies have examined the role of PAX3 in early drug tolerance to BRAF and MEK inhibitors in melanoma [150]. Signaling through BRAF and MEK suppresses PAX3 expression. In melanomas treated with BRAF and MEK inhibitors, PAX3 and MITF expression increases and appears to contribute to the early drug tolerance state [150]. Therefore, inhibitors of PAX3/MITF expression may sensitize melanomas to BRAF/MEK inhibitors. In rhabdomyosarcoma, PAX3 may contribute to apoptotic resistance by reducing PTEN expression [151]. Inducible PAX3 expression in myoblasts downregulates the expression of PTEN, a tumor suppressor that controls progression through the cell cycle by inhibiting G1-S and G2-M transitions [151].

*PAX2* and *PAX8* expressions are also commonly detected in Wilms tumor, a pediatric kidney tumor. One study showed that PAX8 was expressed in over 80% of Wilms tumor cases by immunohistochemistry, while PAX2 was present in over 90% of cases (n = 45) [152]. Interactions between these two PAX genes and *WT1*, a well-known regulator of kidney development and a tumor suppressor in Wilms, have been shown to drive the malignant transformation of nephron progenitor cells [129,153,154]. In normal kidney development, WT1 is able to reduce *PAX2* and *PAX8* expression, driving cells toward a more differentiated state [129,153]. If WT1 is mutated, or its expression is absent or reduced, *PAX2* and *PAX8* expression can continue, maintaining a proliferative, self-renewing state.

### 5.2. PAX Gene Fusions in Cancer

Gene fusions are created as a result of chromosomal translocations and may be capable of producing functional protein products with characteristics shared by their component parts, as well as acquiring neomorphic functions. Several PAX gene fusions have been reported in human cancers.

In lymphomas and leukemias, PAX5 fusions can play major roles in tumorigenesis. In B cell acute lymphoblastic leukemia (B-ALL), a number of PAX5 fusion proteins have been reported, including PAX5::ETV6, a fusion of the paired domain from PAX5 to a large portion of the ETV6 protein, which includes its helix-loop-helix and ETS domains [135]. In non-Hodgkin lymphoma, a translocation juxtaposing the complete coding region PAX5 with IGH is often observed, representing the aberrant upregulation of PAX5 expression [155]. In both cases, these fusion genes have been linked to disease subgroups with poorer prognosis [156,157].

In alveolar rhabdomyosarcoma (ARMS), a chromosomal translocation fusing *FOXO1* to either *PAX3* or *PAX7* is present in about 80% of cases [141,144,158]. A report from the Children’s Oncology Group found that 5-year overall survival was lower in ARMS cases with PAX3::FOXO1 fusions (64%) or PAX7::FOXO1 fusions (87%) than in fusion-negative ARMS (89%) [158]. In these fusions, the paired domain, octapeptide motif, and homeodomain of the PAX protein are retained and connected to a portion of the FOXO1 DNA-binding domain and its entire C-terminal transactivation domain. The exact mechanisms by which these PAX fusions contribute to oncogenesis are not fully understood, but it is known that PAX3::FOXO1 is capable of a 100-fold increase in transcription compared to PAX3 at the regulatory sites of target gene transcription [142]. This increase in transcriptional activity is related to both the loss of the PAX C-terminal transactivation domain as well as the gain of the FOXO1 C-terminal transactivation domain; it appears that while the N-terminal regions of PAX3 are capable of suppressing the transcriptional activity of their own transactivation domain, they cannot suppress transcription by the FOXO1 transactivation domain in the fusion protein [159]. Additionally, it has been shown that PAX3::FOXO1 is capable of interacting with epigenetic coregulators such as CHD4 and BRD4, altering chromatin structure and potentially favoring a self-renewing myogenic state [160,161]. Interestingly, for both PAX5::ETV6 and PAX3/7::FOXO1, a portion of the DNA-binding domain in each of the non-PAX genes is retained in the resulting fusion protein. To our knowledge, the exact functions of these partially retained non-PAX DNA-binding domains in the context of their fusion proteins have not been investigated. The upregulation of PAX3/7::FOXO1 fusion expression relative to wild-type PAX3/7 expression also appears to contribute to their oncogenic effects. There are two distinct mechanisms for the overexpression of each fusion; PAX7::FOXO1 expression is increased by gene amplification, while PAX3::FOXO1 expression is increased via transcriptional upregulation [162].

Lastly, a PAX8::PPARγ fusion is reported in one-third of follicular thyroid carcinomas. Under normal conditions, PPARγ is present at low levels in the thyroid, and the fusion can often be detected simply by looking for an increase in PPARγ staining by immunohistochemistry [163]. PPARγ is itself a transcription factor and is a member of the type II nuclear receptor family. PAX8::PPARγ fusions have been shown to contribute to tumorigenesis via a dominant negative effect, whereby the fusion prevents transcription via the normal PPARγ protein, as well as by normal PAX8 [138,164].

### 5.3. PAX Genes as Favorable Prognostic Indicators

PAX gene expression is not always associated with an unfavorable cancer prognosis. It has been shown that higher expression levels of *PAX1* and *PAX9* are correlated with favorable outcomes in several cancers, as compared to cases with lower expression levels of these genes. In the squamous cell carcinoma of the esophagus, *PAX9* expression is lost as precancerous lesions undergo malignant transformation [127]. Hypermethylation at the *PAX1* promoter and a reduction in *PAX1* expression have also been reported in several different carcinomas, including cervical, ovarian, and colorectal carcinoma, although it is unclear whether these are related to tumor initiation, progression, or maintenance [125,126,165].

In glioblastoma, low *PAX6* expression correlates with unfavorable patient outcomes, and transfecting glioblastoma cells with *PAX6* triggers cell death [147,166]. However, this does not per se make *PAX6* a tumor suppressor. In pancreatic carcinoma, both the canonical PAX6 isoform (a) and isoform (b) (also known as 5a) are expressed and promote the expression of MET, which has been linked to pancreatic cancer progression [146,167]. Therefore, whether a PAX protein functions within an oncogenic pathway may depend on tissue-specific context. Additionally, the high expression of PAX genes in cancer subtypes with a more favorable prognosis does not necessarily indicate that these PAX genes are playing a role in tumor suppression. Heterogeneous oncogenic mechanisms across tumor subtypes can contribute to changes in gene expression that are ultimately related to the oncogenic phenotype.

## 6. Therapeutic Strategies Targeting PAX Genes

The crucial role that PAX genes play in developmental disorders and cancer makes them attractive therapeutic targets, but there are many challenges in designing these potential treatments. One major challenge is that the PAX genes exhibit a high level of intrinsic disorder in their protein structures, which hampers the design of targeted small molecules (Figure 5). In general, transcription factors are more likely to contain regions of intrinsic disorder than non-transcription factor proteins [168]. The percentage of each full-length PAX protein that is disordered ranges from 52.9% in PAX4 to 78.8% in PAX1, as predicted by AlphaFold and curated in MobiDB [169,170,171]. This intrinsic disorder contributes to another complication in targeting PAX proteins; intrinsically disordered proteins can phase separate into membrane-less concentrates. On one hand, this sequestration, coupled with the predominantly nuclear localization of PAX proteins, can make drug delivery more challenging. On the other hand, delivery to these phase-separated condensates could increase local drug concentrations and help achieve the therapeutic effect. Drug design is complicated by the fact that PAX genes may play different roles in different tissue types and at different developmental time points. Despite these challenges, several groups have designed PAX-directed treatments for a variety of cancers. These will be discussed below, along with potential future directions for drug design in PAX-related diseases.

### 6.1. Indirect Targeting of PAX Proteins

Several groups have reported drugs capable of inhibiting PAX partner proteins or of phenocopying the loss of PAX gene expression. An indirect inhibitor of PAX2 called BG-1 has recently been reported and has been shown to prevent interactions between PAX2 and histone methyltransferases in renal cell cultures [172]. In particular, the authors employed a cell-based strategy in which drugs were tested for their ability to inhibit luciferase signaling from a transfected PAX2-responsive reporter. This drug was also capable of inhibiting the proliferation of renal carcinoma cells.

Several indirect inhibitors of the PAX3::FOXO1 fusion protein have also been reported. For example, entinostat, an HDAC inhibitor, is capable of inhibiting PAX3::FOXO1 at the transcriptional level, as well as the growth of fusion-positive xenografts in mice [173]. Inhibiting interactions between PAX3::FOXO1 and its protein-binding partners CHD4 and BRD4 has resulted in phenotypes that resemble PAX3::FOXO1 depletion [160,161]. SAHA, a histone deacetylase inhibitor, and fenretinide, a vitamin A analog, have also been shown to reduce PAX3::FOXO1 protein levels in rhabdomyosarcoma cells [174,175]. In the above examples, no drug binds directly to the PAX3::FOXO1 protein. Rather, these drugs reduce PAX3::FOXO1 expression via epigenetic deregulation, the disruption of protein–protein interactions, and/or the direct transcriptional suppression of PAX3::FOXO1.

### 6.2. Direct Targeting of PAX Proteins

The direct targeting of PAX proteins is challenging, as they are transcription factors with high levels of intrinsic disorder in their protein structures. One strategy is to employ structure-based virtual screening, which has been used with success in discovering EG1, a drug capable of inhibiting the DNA-binding capability of PAX2 [176]. In this instance, the authors followed up in silico predictions with biolayer interferometry (BLI), a label-free method that detected binding between PAX2 and EG1 with a binding affinity of 1.35 µM. EG1 was also capable of inhibiting the transcriptional activity of PAX2, PAX5, and PAX8 in cell-based activities using luciferase reporters, as well as being able to inhibit embryonic kidney development.

Another potential strategy for the detection of potential PAX inhibitors is to use an in vitro drug screen upfront, such as BLI or surface plasmon resonance, to detect drugs capable of binding directly to immobilized recombinant PAX protein, which could then be narrowed to potential inhibitors using luciferase screens and functional assays. This strategy has recently been employed to identify piperacetazine, a first-generation antipsychotic, as a compound capable of binding to PAX3::FOXO1 and inhibiting the protein’s transcriptional activity [177].

### 6.3. Additional Strategies

To combat the difficulty of finding direct inhibitors of PAX proteins, another potential strategy could be the employment of oligonucleotide-based therapies such as antisense oligonucleotides (AONs) or RNA interference. In vitro, AONs targeting *PAX3* have been shown to inhibit PAX3::FOXO1 expression and trigger apoptosis in a fusion-positive rhabdomyosarcoma cell line [178]. Unfortunately, issues with the delivery and safety of oligonucleotide-based therapies currently limit their practical application. However, oligonucleotide-based therapeutics are undergoing rapid development and improvement. In the past ten years, the FDA has approved over a dozen oligonucleotide-based therapeutics. Prior to that timeframe, only two oligonucleotide-based drugs, fomiversen (1998) and pegaptinib (2004), had gained FDA approval. Several chemical modifications have improved the stability and delivery of oligonucleotide drugs. First-generation modifications include alterations to the phosphate backbone [179]. Second-generation modifications alter the 2′-OH to improve stability and reduce nuclease degradation [179]. Third-generation modifications include morpholino oligomers that replace phosphodiester bonds with phosphorodiamidate bonds and protein nucleic acids that replace phosphodiester bonds with amide bonds [179]. Conjugation to other molecules, such as polyethylene glycol (PEG), has also been employed to improve the pharmacokinetic properties of these drugs [179].

Another potential strategy that could be of use in the treatment of tumors with PAX fusions is immunotherapy. In particular, cancer vaccines could be developed to target the breakpoints of these fusions, as they would not be present in healthy cells. To date, one group has discovered a PAX3::FOXO1 breakpoint peptide capable of inducing circulating T lymphocytes to lyse tumor cells. While further studies demonstrated that an immune response could be generated in patients with fusion-positive rhabdomyosarcoma, this response was neither long-lasting nor consistent, and additional improvements would be needed prior to clinical use [180].

## 7. Conclusions

PAX family members have been recognized as key mediators of human development and disease for decades, but recent advances in our understanding of these underlying biological processes, as well as technological advances in drug development, have made PAX-specific therapies an emerging possibility. Continued progress will require a detailed understanding of the differences between the PAX genes, the functional consequences of structural changes and PAX isoforms, and a broader knowledge of the cellular and tissue-specific environments in which these proteins exert their effects.

Future studies could focus on the roles of PAX genes when expressed in adult tissues, particularly in the roles of these genes in regeneration after tissue injury. It will also be necessary to explore differences in the timing of expression, tissue localization, and functional roles of PAX protein isoforms. Further research is also needed in the functional consequences of specific mutations found in PAX genes in developmental disorders. Lastly, the roles that wild-type PAX gene expression may have in different cancer types have still not been fully elucidated, and PAX genes may support tumor initiation or maintenance, act as tumor suppressors, or be expressed as a consequence of other oncogenic processes without having a direct oncogenic effect.

## Figures and Tables

**Figure 1 cancers-16-01022-f001:**
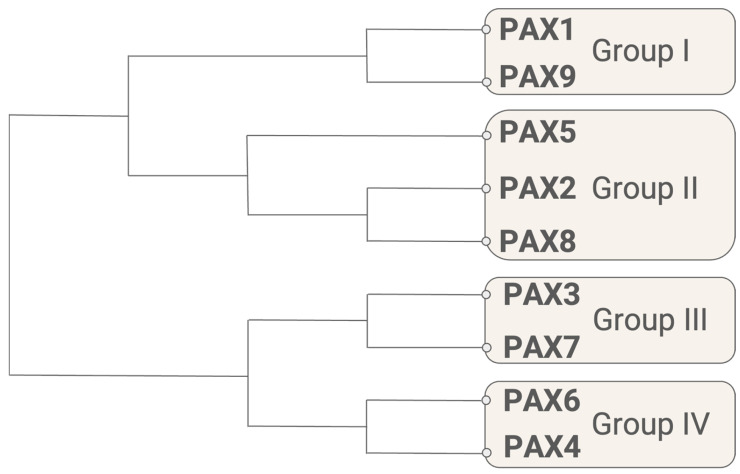
Phylogenetic tree of human PAX proteins. The amino acid sequences aligned for this tree are NCBI reference sequences for the longest isoform of each protein (Table 1). Alignment and phylogenetic reconstructions were performed using the function “build” of ETE 3.1.2 as implemented on the GenomeNet “https://www.genome.jp/tools/ete/ accessed on 20 July 2023”. [7]. Alignment was performed using MAFFT v6.861b with the default options [8]. ML tree was inferred using IQ-TREE 1.5.5 run with ModelFinder and tree reconstruction [9]. Tree branches were tested using SH-like aLRT with 1000 replicates.

**Figure 2 cancers-16-01022-f002:**
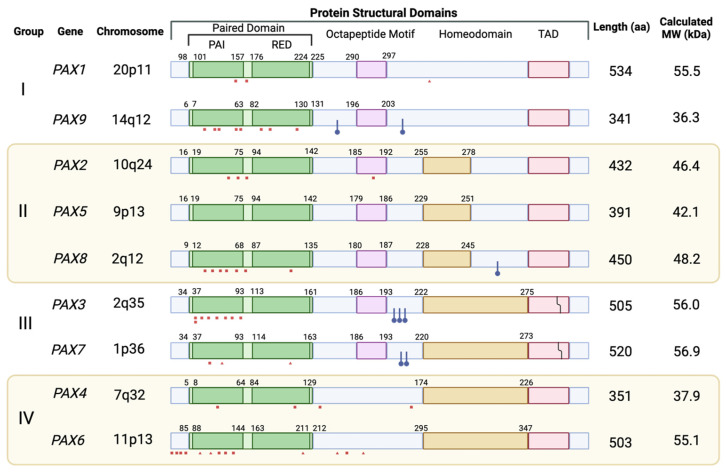
Human PAX subgroups by protein structure. All nine human PAX proteins include the N-terminal paired domain, which contains two helix-turn-helix motifs that mediate binding to DNA. The octapeptide linker motif found in I, II, and III is typically associated with transcriptional repression mediated by the binding of other regulatory factors. Group II PAX genes possess a partial version of the homeodomain with a reduced DNA-binding capability. The full-length homeodomain is present in Groups III and IV and contains a helix-turn-helix motif that dimerizes for DNA binding. All nine family members also possess a C-terminal transactivation domain. Red squares indicate missense mutations associated with developmental diseases. Red triangles indicate nonsense mutations associated with developmental diseases. Blue dots indicate confirmed phosphorylation sites. Jagged lines indicate breakpoints for the PAX3::FOXO1 and PAX7::FOXO1 fusion proteins observed in alveolar rhabdomyosarcoma. For each protein, the longest isoform is depicted. This figure is not to scale.

**Figure 3 cancers-16-01022-f003:**
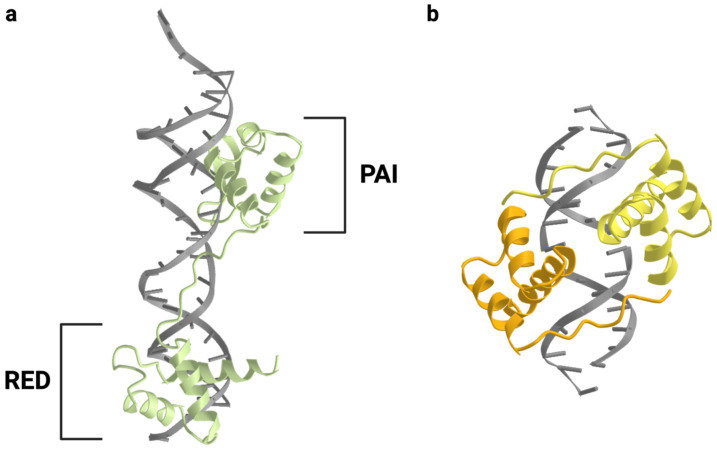
Structure of the paired domain and homeodomain. (**a**) In PAX proteins, the paired domain consists of two separate helix-turn-helix motifs, the PAI and RED subdomains, connected by a flexible linker. Here, the paired domain of human PAX6 is depicted bound to DNA [15]. (**b**) PAX homeodomains bind DNA as dimers. A human PAX3 homeodomain dimer is depicted bound to DNA [18]. Structural representations were created using the Molecular Modeling Database (MMDB) hosted by the NCBI [20].

**Figure 4 cancers-16-01022-f004:**
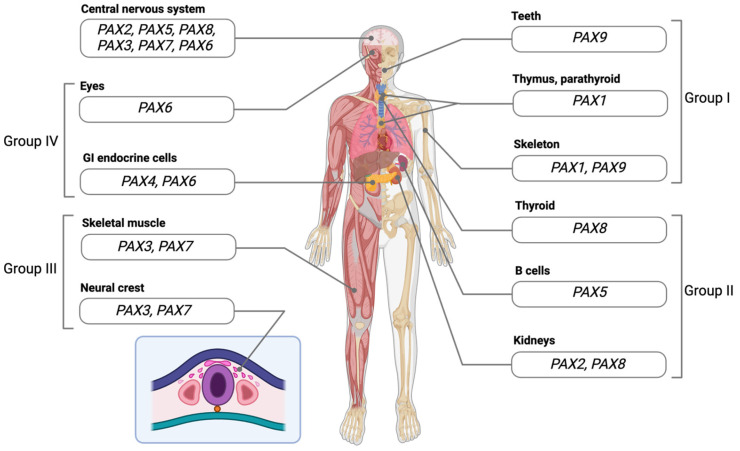
Developmental roles of PAX genes. PAX genes play multiple, often overlapping roles in the development of human tissue types. Examples of cell types/tissues/organs regulated by PAX genes are depicted here and are also listed in Table 2.

**Figure 5 cancers-16-01022-f005:**
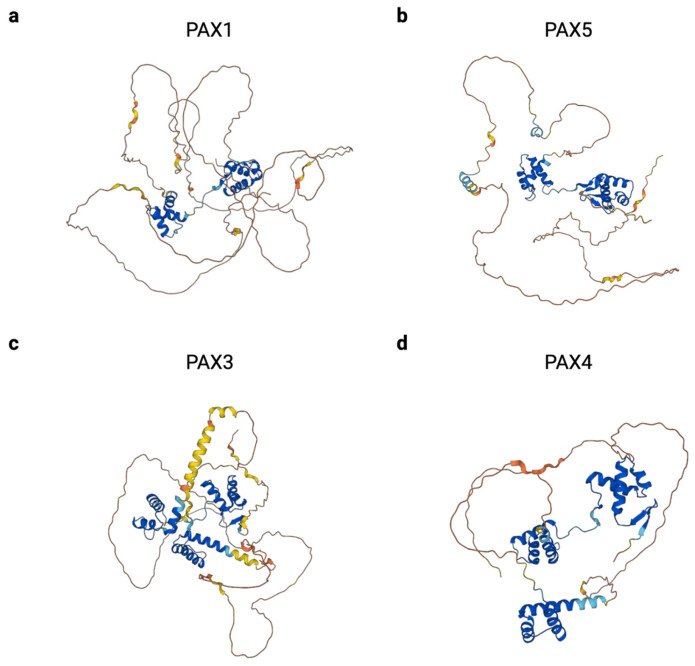
Predicted 3D structures of PAX proteins exhibit large regions of intrinsic disorder. The structure of one representative protein from each PAX group was rendered by AlphaFold [169,170]. AlphaFold produces a per-residue confidence score (pLDDT) between 0 and 100. Model confidence is indicated as follows. Very high: dark blue (pLDDT > 90), confident: light blue (90 > pLDDT > 70), low: yellow (70 > pLDDT > 50), very low: orange (pLDDT < 50). (**a**) Group I: PAX1 (**b**) Group II: PAX5 (**c**) Group III: PAX3 (**d**) Group IV: PAX4.

**Table 1 cancers-16-01022-t001:** Human PAX isoforms. This table summarizes the human PAX protein isoforms. The longest and shortest isoforms are listed to give a sense of the variation in isoform sizes within and between groups. Not all isoforms are detected in adult human tissues. The list of isoforms was generated using all listed protein isoforms from the NCBI RefSeq entry for the human version of each gene. Isoforms are listed by their names in the NCBI RefSeq database.

Subgroup	Gene	Isoforms	Longest Isoform (# of Amino Acids)	Shortest Isoform (# of Amino Acids)
Group I	*PAX1*	1, 2	1 (534)	2 (457)
	*PAX9*	-	341	-
Group II	*PAX2*	a–g (7)	e (432)	g (102)
	*PAX5*	1–11	1 (391)	6 (220)
	*PAX8*	A, C, D, E (4)	A (450)	E (287)
Group III	*PAX3*	a, b, c, d, e, g, h, i	e (505)	b (206)
	*PAX7*	1–3	1 (520)	3 (505)
Group IV	*PAX4*	1, 2	1 (351)	2 (348)
	*PAX6*	a–o (15)	e (503)	o (221)

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
