# Peer review of "The PAX Genes: Roles in Development, Cancer, and Other Diseases"

_cancers, 2024, doi:10.3390/cancers16051022_

Round 1

Reviewer 1 Report

Comments and Suggestions for Authors

The review entitled “The PAX Genes: Roles in Development and Disease” provides a broad perspective on PAX gene structure and function, and narrates how the PAX genes may be altered in disease pathogenesis.

Paired-box (PAX) genes encode a family of highly conserved transcription factors found in vertebrates and invertebrates. PAX proteins orchestrate tissue and organ development throughout cell differentiation and lineage determination and often termed "master regulators". Mutations in PAX genes are associated with different diseases including cancers. Several reviews on this are available in literature. The current review narrates in-depth analysis on PAX genes including evolution of PAX-like genes, PAX protein structure, regulation of PAX family gene expression, PAX genes during development, PAX genes in human disease and PAX genes and cancer.  The topic of the review is very interesting and the writing is very good. The review can be a good reference literature of researchers working on the field.

I would like to request the authors to briefly narrate the future research areas on the topic. Overall, the review is very good and I do not have much concern.

Author Response

Thank you for your feedback!

I would like to request the authors to briefly narrate the future research areas on the topic.

At your suggestion, we have added some potential future research areas involving PAX genes to the Conclusions section of the paper (lines 788-797).

At the suggestion of the editors, we have also changed the title of this manuscript to "The PAX Genes: Roles in Development, Cancer and Other Diseases."

Reviewer 2 Report

Comments and Suggestions for Authors

PAX gene encode transcription factors which regulate basic developmental processes. This well known group of genes has been studied over decades. Accordingly, a huge amount of papers have been published about this topic. Nevertheless, the authors managed to write a comprehensive, focused and well written review.

However, I have found several minor points which require correction:

1. Figure 1 shows the result of an alignment of human PAX protein sequences which generates four groups. This kind of grouping is not new. Please cite and mention according papers which show the same result. The methods used for this alignment are cited not according to their appearance in the text.

2. The legends of Table 1 and 2 are mixed up. Please correct additionally the numbering of these tables in the text.

3. Table 1 shows PAX isoforms. Please indicate the source of these data. Add citations or indicate the method of their generation.

4. In table 2 you indicate no data for PAX5. I would prefer to add here the information about leukemia/lymphoma although mentioned later. But cancer is also a disorder. Please mention in the legend that the literature is mentioned in the text.

5. Please delete the sentence in line 154-155.

6. Please clarify the relation of the genes Ref-1 and APEX1 (line 220). I think that they are identical.

7. In the section PAX genes during development you introduce the term master regulators and give the Yamanaka factors as example. This is no adequate example in my eyes. These factors are responsible to generate/maintain a progenitor status of cells. Master factors are driving the differentiation of specific cell types/tissues/organs. Better non-PAX expamples at this point are homeodomain factors NKX2-5 (driving development of the heart) or NKX3-1 (driving the development of spleen).

8. Please change in the legend of Figure 4 the part "Some of the key developmental programs .." into "Examples of cell types/tissues/organs ..".

9. PAX5 is not described in the section PAX genes in human disease. Please add some information at this point.

10. Please add literature for the statements that PAX factors exhibit high levels of intrinsic disorder (lines 679 and 682).

Author Response

Thank you for your feedback! We have made the following changes to address your comments:

  1. Figure 1 shows the result of an alignment of human PAX protein sequences which generates four groups. This kind of grouping is not new. Please cite and mention according papers which show the same result. The methods used for this alignment are cited not according to their appearance in the text.

We have included references to the publication that first generated the PAX gene groupings used here, and have updated the citations to match order of appearance in the text (lines 55-57, Figure 1 caption).

  1. The legends of Table 1 and 2 are mixed up. Please correct additionally the numbering of these tables in the text.

Legends and table numbering have been corrected (Tables 1 and 2).

  1. Table 1 shows PAX isoforms. Please indicate the source of these data. Add citations or indicate the method of their generation.

A list of protein isoforms was generated from the entries for the human form of each gene on NCBI's RefSeq database (Table 1 legend).

  1. In table 2 you indicate no data for PAX5. I would prefer to add here the information about leukemia/lymphoma although mentioned later. But cancer is also a disorder. Please mention in the legend that the literature is mentioned in the text.

We have indicated in the legend that data regarding each gene's involvement in human cancers will be discussed later. We have also indicated that references for each disorder can be found in the text (Table 2 legend).

  1. Please delete the sentence in line 154-155.

This sentence has been removed.

  1. Please clarify the relation of the genes Ref-1 and APEX1 (line 220). I think that they are identical.

You are correct. This section has been rephrased for clarity (lines 220-226).

  1. In the section PAX genes during development you introduce the term master regulators and give the Yamanaka factors as example. This is no adequate example in my eyes. These factors are responsible to generate/maintain a progenitor status of cells. Master factors are driving the differentiation of specific cell types/tissues/organs. Better non-PAX expamples at this point are homeodomain factors NKX2-5 (driving development of the heart) or NKX3-1 (driving the development of spleen).

This section has been updated with Nkx2-5 as an example of a master regulator of the heart (lines 256-264).

  1. Please change in the legend of Figure 4 the part "Some of the key developmental programs .." into "Examples of cell types/tissues/organs ..".

This sentence has been changed as suggested (Figure 4 legend).

  1. PAX5 is not described in the section PAX genes in human disease. Please add some information at this point.

We chose to focus this section on non-cancer diseases, specifically on diseases listed in the OMIM database under the entry for each PAX gene. For PAX5, no non-cancer diseases were listed in the OMIM database. Acute lymphoblastic leukemia was listed, and is included in the section on PAX genes in cancer (Table 3, paragraph beginning on line 630).

  1. Please add literature for the statements that PAX factors exhibit high levels of intrinsic disorder (lines 679 and 682).

This statement regarding the intrinsic disorder of PAX proteins has been edited to indicate the predicted percentage of disorder as a range across all PAX proteins, as determined using AlphaFold. We have also included a more general citation about higher levels of intrinsic disorder observed in transcription factors (lines 697-700).

At the suggestion of the editors, we have also changed the title of this manuscript to "The PAX Genes: Roles in Development, Cancer and Other Diseases."